# Knockout of *myoc* Provides Evidence for the Role of Myocilin in Zebrafish Sex Determination Associated with Wnt Signalling Downregulation

**DOI:** 10.3390/biology10020098

**Published:** 2021-01-30

**Authors:** Raquel Atienzar-Aroca, José-Daniel Aroca-Aguilar, Susana Alexandre-Moreno, Jesús-José Ferre-Fernández, Juan-Manuel Bonet-Fernández, María-José Cabañero-Varela, Julio Escribano

**Affiliations:** 1Área de Genética, Facultad de Medicina de Albacete/Instituto de Investigación en Discapacidades Neurológicas (IDINE), Universidad de Castilla-La Mancha, 02006 Albacete, Spain; Raquel.Atienzar@uclm.es (R.A.-A.); JoseDaniel.Aroca@uclm.es (J.-D.A.-A.); Susana.Alexandre@uclm.es (S.A.-M.); ferrejesus@hotmail.com (J.-J.F.-F.); JuanM.Bonet@uclm.es (J.-M.B.-F.); mjosefa.cabanero@uclm.es (M.-J.C.-V.); 2Cooperative Research Network on Age-Related Ocular Pathology, Visual and Life Quality (OFTARED), Instituto de Salud Carlos III, 28029 Madrid, Spain

**Keywords:** myocilin, *myoc*, zebrafish sex determination, Wnt

## Abstract

**Simple Summary:**

Myocilin is a protein with an incompletely understood function, mainly known because of its role in glaucoma. In this study we have analysed the normal role of this protein in vivo. To that end, we generated the first myocilin knockout zebrafish line reported to date. This zebrafish line did not show any apparent gross morphological anomaly, but unexpectedly, we observed that all knockout animals were males. Detailed analyses revealed the existence of apoptosis in the immature juvenile gonad, which is associated with male differentiation. Moreover, we demonstrate that adult knockout differentially expressed key genes involved both in male sex determination and the Wnt signalling pathway, which also plays a role in zebrafish gonad differentiation. Altogether, these results indicate that myocilin is a novel key protein involved in sex determination in zebrafish.

**Abstract:**

Myocilin is a secreted glycoprotein with a poorly understood biological function and it is mainly known as the first glaucoma gene. To explore the normal role of this protein in vivo we developed a *myoc* knockout (KO) zebrafish line using CRISPR/Cas9 genome editing. This line carries a homozygous variant (c.236_239delinsAAAGGGGAAGGGGA) that is predicted to result in a loss-of-function of the protein because of a premature termination codon p.(V75EfsX60) that resulted in a significant reduction of *myoc* mRNA levels. Immunohistochemistry showed the presence of myocilin in wild-type embryonic (96 h post-fertilization) anterior segment eye structures and caudal muscles. The protein was also detected in different adult ocular and non-ocular tissues. No gross macroscopic or microscopic alterations were identified in the KO zebrafish, but, remarkably, we observed absence of females among the adult KO animals and apoptosis in the immature juvenile gonad (28 dpf) of these animals, which is characteristic of male development. Transcriptomic analysis showed that adult KO males overexpressed key genes involved in male sex determination and presented differentially expressed Wnt signalling genes. These results show that myocilin is required for ovary differentiation in zebrafish and provides in vivo support for the role of myocilin as a Wnt signalling pathway modulator. In summary, this *myoc* KO zebrafish line can be useful to investigate the elusive function of this protein, and it provides evidence for the unexpected function of myocilin as a key factor in zebrafish sex determination.

## 1. Introduction

Myocilin is a 55–57 kDa extracellular glycoprotein with an enigmatic function and was identified in 1997 as the first glaucoma gene [1]. Glaucoma is a progressive and irreversible optic neuropathy that is caused by apoptosis of retinal ganglion cells, and it is generally associated with elevated intraocular pressure [2]. This protein was identified in human trabecular meshwork cell cultures that were treated with glucocorticoids, and it was initially called Trabecular Meshwork Inducible Glucocorticoid Response (TIGR) [3]. It was later called myocilin because of its amino acid sequence similarity with myosin [4]. The transcripts encoding myocilin were initially discovered in the ciliary body [5,6] and then in photoreceptor cells [4]. The gene is also expressed in other tissues of the ocular anterior segment such as the iris and the trabecular meshwork (TM) [6,7,8,9]. The protein is present in the aqueous humor [10,11], where it forms aggregates of 120–180 kDa [12], which are linked partially by disulphide bonds [13]. In addition, *MYOC* expression has been detected in non-ocular tissues such as skeletal and cardiac muscles [6], blood plasma, leukocytes and lymphoid tissues [14]. The protein has been reported to be secreted in association with exosomes in TM cells [15,16].

Although important structural properties of myocilin have been unveiled, they have not provided a definitive clue to elucidate its normal function. Thus, we know that the N-terminal region of myocilin is composed of two coiled-coil domains [17,18] with a leucine-zipper motif [6] in the second coiled-coil, which are involved in myocilin self-aggregation [12]. The N-terminal half is connected to the C-terminal part of the protein by a central region that contains a calpain II proteolytic site that is cleaved intracellularly [11,19]. The C-terminal region is homologous to olfactomedin [6], and identifies this protein as a member of the olfactomedin protein family. This family comprises a group of glycoproteins that are known to be involved in early development and functional organisation of the nervous system as well as haematopoiesis. Olfactomedin domains appear to facilitate protein–protein interactions, intercellular interactions and cell adhesion [20]. The olfactomedin domain of myocilin folds like a globular five-bladed β-propeller [21] and contains most glaucoma-causing variants [6]. The quaternary structure of myocilin is composed of a Y-shaped dimer-of-dimers in which the N-terminal coiled-coil region forms a tetrameric stem that is linked by disulphide bonds, and it is connected through the linker region to two pairs of olfactomedin domains [22]. We have proposed that extracellular myocilin may form a dynamic extracellular network that is composed of myocilin homoaggregates which may bind through the olfactomedin domain with matricellular proteins such as SPARC and hevin, as well as fibronectin, which suggests that myocilin might function as a putative matricellular protein [23,24].

Myocilin and other olfactomedin family members, such as photomedin-1 [25], gliomedin [26] and latrophilin [27] are proteolytically cleaved, splitting the proteins in two fragments. Though the role of this process is not completely understood, we have proposed that it regulates molecular interactions of this protein [23,24]. We have shown that it is affected by the extracellular concentration of bicarbonate [28]. C-terminal myocilin fragments have been identified in different ocular tissues and biological fluids such as the ciliary body, aqueous humor (AH) [11] and trabecular meshwork [9], indicating that the proteolytic cleavage of this protein also occurs in vivo and that it may be important in regulating its biological function. Interestingly, recombinant myocilin has been reported to modulate Wnt signalling in cell culture, suggesting that this pathway might also participate in its normal function [29,30]. Myocilin has anti-adhesive properties on trabecular meshwork cells [31,32] and reduces the adhesion of human circulating leukocytes to cultured endothelial cell monolayers [14].

The sex determination mechanisms of zebrafish remain largely unknown. Wild zebrafish strains use a ZZ/ZW genetic sex determination process, with the major sex locus located on chromosome 4. This genetic sex determinant seems to have been lost in domesticated strains, which are widely used in the laboratory [33]. Rather than a single master gene, multiple genes and possible weak secondary environmental factors have been proposed to play a role as sex determining factors in zebrafish research strains [34]. Environmental factors that are known to influence zebrafish sex include temperature, density, hormones, food and hypoxia [34]. The zebrafish gonad, like the mammalian gonad, is bipotential before sex determination (<10 dpf), and it is composed of a mixture of male- and female-like cells [35,36]. Sex differences begin to be apparent when the number of oocytes tend to increase in females (20–30 dpf) [37]. This process is regulated by canonical Wnt/beta-catenin signalling, which acts as a pro-female pathway [38], and inhibition of this pathway results in male-biased sex ratios.

In this study, we developed a *myoc* knockout (KO) zebrafish model to explore the biological role of this interesting protein. We found that while this line did not show apparent morphological anomalies, unexpectedly, all KO animals were males, indicating that myocilin is required for ovary differentiation, acting as a novel key protein involved in sex determination in domesticated zebrafish. To the best of our knowledge, our results also provide the first in vivo evidence of myocilin as a Wnt signalling pathway modulator.

## 2. Materials and Methods

### 2.1. Animals

Wild-type AB zebrafish (*Danio rerio*) were maintained at 28 °C with a 14 h on/10 h off light cycle and were fed a standard diet according to established protocols [39]. Zebrafish embryos were raised at 28 °C in E3 medium (5 mM NaCl; 0.17 mM KCl; 0.33 mM CaCl_2_; 0.33 mM MgSO_4_ and 0.0001 % methylene blue, pH 7.2). Adult fishes and larvae were anesthetized with 0.04% and 0.02% tricaine methanesulfonate (MS222, Sigma-Aldrich, St. Louis, MO, USA), respectively.

### 2.2. CRISPR/Cas 9 Gene Editing

Target selection and crRNA design were performed using custom Alt-R CRISPR-Cas9 guide RNA (https://eu.idtdna.com/site/order/designtool/index/CRISPR_CUSTOM, Integrated DNA Technologies, Coralville, IA USA). Potential off-target sites and highest on-target activity of crRNAs were assessed with CRISPR-Cas9 guide RNA design checker (https://eu.idtdna.com/site/order/designtool/index/CRISPR_SEQUENCE, Integrated DNA Technologies). TracrRNA and crRNA targeting *myoc* exon 1 (myocE1g1 5’-GGTTGCTCGTCTCGTAGGAGGGG-3’) were purchased from Integrated DNA Technologies. For Cas9/gRNA microinjections, crRNA (36 ng/µL) and tracrRNA (67 ng/µL) were mixed, incubated 5 min at 95 °C and cooled at room temperature to hybridise. Cas9 protein (Alt-R^®^ CRISPR-Cas9 at 250 ng/µL, Integrated DNA Technologies) and crRNA/tracrRNA complex were mixed and incubated for 10 min at 37 °C to form the RNP complex. Approximately 3 nl of RNP complex was injected into the animal pole of one-cell stage embryos (50–250 embryos/experiment) using a Femtojet 5247 microinjector (Eppendorf, Hamburg, Germany) under a Nikon DS-Ri2 stereomicroscope. As a negative control, embryos were injected with Cas9/tracrRNA and no crRNA.

### 2.3. Zebrafish DNA Extraction

PCR-ready genomic DNA was isolated from whole zebrafish embryos (24 h post fertilization, hpf) and from the caudal fin of anesthetized larvae (144 hpf) or adult zebrafish using the HotSHOT method [40]. Briefly, tissue samples were incubated with 20 µL of base solution (25 mM KOH, 0.2 mM EDTA) at 95 °C for 30 min in a thermal cycler (BIORAD C100, BIORAD, Hercules, CA, USA), then 20 µL of neutralization buffer (40 mM TrisHCl, pH 5) was added.

### 2.4. Genotyping of CRISPR/Cas9-Induced Mutations by PAGE and Sanger Sequencing

To characterize the KO mutation, *myoc* exon 1 was amplified by PCR in a thermal cycler (BIORAD C100) using the following conditions: an initial denaturation step at 95 °C for 3 min followed by 35 cycles consisting of denaturation at 95° C for 30 s, annealing at 66 °C for 30 s and extension at 72° C for 30 s. A final extension step at 72° C for 5 min was also included. The primers (*myoc*Fw1, 5’-GGTCGCTGTCAGTACACCTTTAT-3’; *myoc*Rv1, 5’-GCAGGTCCTGAACTTGTCTGTCT-3’) were designed using the IDT PrimerQuest Tool (https://eu.idtdna.com/Primerquest/Home/Index, Integrated DNA Technologies) and the PCR products were analysed either by DNA PAGE (8%) or direct Sanger sequencing (Macrogen, Seoul, Korea). PAGE was carried out using the Mini-PROTEAN III gel electrophoresis system (BioRad). After electrophoresis, the gel was stained for 20 min in a Ethidium Bromide (46067, Sigma-Aldrich, St. Louis, MO, USA) solution (0.5 µg/L).

### 2.5. Quantitative Reverse Transcription PCR (qRT-PCR)

qRT-PCR was carried out as previously described [41]. RNA was isolated from pools of 15 zebrafish larvae (144 hpf) or from pools of three adult male zebrafish (2.5 months) using the RNeasy Minikit (#74104, Qiagen, Germantown, MD, USA) and treated with RNase-free DNase I according to the manufacturer’s instructions. Purified RNA was used for cDNA synthesis using RevertAid First Strand cDNA Synthesis Kits (#K1622, Thermo Fisher Scientific, Waltham, MA, USA). The expression of *myoc* mRNA or of selected DEGs relative to *ef1α* mRNA was determined using the 2^−ΔΔCt^ method [42] using the primer pairs described in Appendix A. The PCR analysis was carried out with 1 μL of cDNA as a template in a reaction volume of 10 μL containing 5 μL of Power SYBR Green PCR Master Mix (Thermo-Fisher Scientific) and 200 nM of each primer. Thermocycling conditions included an initial denaturation step at 95° C for 10 min, followed by 40 cycles consisting of 15 s denaturation at 95° C for 60 s and a combined annealing and extension step at 60 °C for 40 s. The PCR products and their dissociation curves were detected with a 7500 Fast real-time PCR system thermal cycler (Thermo-Fisher Scientific). The template cDNA was omitted in the qRT-PCR negative control. qRT-PCR results from three independent experiments were used for calculation of mean expression values in each sample.

### 2.6. Zebrafish Tissue Samples

For histological sections, wild-type and KO *myoc* zebrafish whole 96 hpf-embryos or adult zebrafish heads were fixed overnight in 4% PFA and cryoprotected two days at 4 °C in 30% sucrose/PBS 0.1 M (Dulbecco, X0515-500C). Thereafter, the embryos and zebrafish heads were embedded in 10% porcine gelatin with 15% sucrose and stored at −80 °C. Serial cryosections (10 μm for embryos and 14 μm for larvae and adult zebrafish) were obtained in a Leica CM3050 S cryostat (Leica Ltd., Wetzlar, Germany) and stored at −20 °C for further use.

### 2.7. Fluorescent Whole Mount Immunohistochemistry (FWIHC)

Phenylthiourea-treated and fixed embryos (96 hpf) were incubated with a chicken primary antibody (1:50) raised against a N-terminal peptide of the human myocilin protein (anti-TNT) [14], followed by incubation with a Cy2 goat anti-chicken IgY (1:1000) secondary antibody (703-225-155, Jackson ImmunoResearch, West Grove, PA, USA). Whole embryos were counterstained with DAPI (D8417, Sigma-Aldrich), mounted in low-melting agarose (1%) (8050, Pronadisa) with Fluoroshield Medium (F6182, Sigma-Aldrich) and visualized in an LSM710 Zeiss confocal microscope. Fluorescence emitted by DAPI, the Cy2-conjugated antibody and embryo autofluorescence was registered at the following wavelengths, respectively: 411–464 nm, 490–518 nm and 553–677 nm. Z-stacks were captured with sections spanning the entire embryo and maximum intensity projections and cross-sections of the confocal images were obtained with ZEN software (Carl Zeiss, Jena, Germany).

### 2.8. Fluorescence Immunohistochemistry

Fluorescence immunohistochemistry was performed as previously described [43,44]. Briefly, gelatin embedded histological sections of juvenile (28 dpf) or adult zebrafish (7 months) were treated with immunoblocking solution [10% fetal bovine serum (FBS), 1% DMSO and 1% Triton X-100 in DPBS] at room temperature for 1 h. Gonadal tissue and myocilin were identified using an antibody against the germ cell marker vasa (1:200 [45], GTX128306, GeneTex, Hsinchu City, Taiwan) or the anti-myocilin (TNT, 1:150) antibody, followed, respectively, by incubation with a Cy2-conjugated anti-chicken IgY (1:1000) or anti-rabbit IgG secondary antibody (1:1000). After that, sections were incubated with a secondary antibody, counterstained, mounted and visualized as described earlier. The specificity of the anti-myocilin antibody was evaluated by incubation with the preimmune antibody (1:200) and with a competitive assay using the antigenic peptide at a 1:5 (antibody:peptide) molar ratio. 

Apoptotic cell death was evaluated by TUNEL assay using the In-Situ Cell Death Detection Kit, Fluorescein (11684795910, Roche Diagnostics, Mannheim, Germany), following the manufacturer’s instructions. As a positive control, tissue sections of wild-type zebrafish were incubated for two min with permeation solution (0.1% Tritón-X100, 0.1% sodium citrate) followed by incubation with DNase I solution (3 U/mL DNase, 50 mM Tris-HCl pH 7.5, 1 mg/mL FBS) for 10 min [41]. DNase I treatment was omitted in the negative controls. Samples were stained with DAPI, mounted and visualized as described earlier. At least four animals from each experimental group were used for the microscopy analyses. Four tissue sections per fish were employed for each technique and three random fields per tissue section were examined by a single masked observer.

### 2.9. Hematoxilin and Eosin Staining

Tissue sections, previously washed in PBS, were stained with Harris hematoxylin solution (HHS80, Sigma-Aldrich) for 3 min. Then, sections were washed with water, dehydrated in ascending ethanol concentrations (15%, 30%, 50%, 70% and 90%) and stained with an alcoholic eosin solution (HT1101116-500ML, Sigma-Aldrich) for 2 min. Finally, the samples were again treated with increasing concentrations of ethanol (90 and 100%) before a final xylol wash at room temperature. Slides were then mounted with Cytoseal (8311-4, Thermo Scientific, Waltham, MA, USA).

### 2.10. High Throughput RNA Sequencing

RNA was isolated from pools of three adult male zebrafish (2.5 months) using the RNeasy Minikit (Qiagen #74104) and treated with RNase-free DNase I according to the manufacturer’s instructions. RNA concentration was determined using a NanoDrop 2000 (Thermo Fisher Scientific). Duplicates of RNA samples were submitted to Macrogen Next Generation Sequencing Division (Macrogen) for high throughput sequencing. Libraries were generated using the TruSeq Stranded mRNA LT Sample Prep Kit (Illumina, Foster City, CA, USA), which captures both coding RNA and multiple forms of non-coding RNA that are polyadenylated. Sequencing was performed in a NovaSeq 6000 System (Illumina) according to the user guide (Document #1000000019358 v02). Trimmomatic 0.38 [46] was used to remove the Illumina adapter sequences and bases with base quality lower than three from the ends. HISAT2 aligner [47] was used to map sequence reads against the zebrafish genome reference (GRCz11). Expression profiles were calculated for each sample as read count and normalization value, which is based on transcript length and depth of coverage. DEG analysis of the *myoc* KO vs. wild-type zebrafish was performed using reads per kilobase of transcript per million mapped reads (RPKM). Genes with a fold change ≥2.0 and a *p*-value < 0.05 in the four possible comparisons of the two biological replicas were considered as DEGs (KO1 vs. WT1, KO1 vs. WT2, KO2 vs. WT1 and KO2 vs WT2). Functional gene enrichment analysis of DEGs was performed using GO (http://geneontology.org/), KEGG (http://www.kegg.jp/kegg/pathway.html) databases using the g:Profiler tool (https://biit.cs.ut.ee/gprofiler/) and ShinyGO [48]. 

### 2.11. Statistics

Statistical comparisons between groups were performed using the chi-squared, Fisher’s tests or one-way ANOVA. Statistical analysis of the data was performed using the SigmaPlot 12.0 software (Systat Software Inc., San Jose, CA, USA).

## 3. Results

### 3.1. Generation of the KO Myoc Line in Zebrafish Using CRISPR/Cas9

To better understand the biological function of *MYOC* in vivo, we used the zebrafish as an animal model. As mentioned above, the human *MYOC* gene consists of three exons and it is located on the long arm of chromosome 1, whereas the orthologue zebrafish gene has four exons and it is located on chromosome 20 (Appendix A). Comparison of DNA sequences showed 37.2% nucleotide identity in the coding regions of the human and zebrafish genes. The corresponding proteins present conserved olfactomedin domains with 45% amino acid sequence identity (Appendix A). The N-terminal coiled-coils that are present in the human protein were not predicted in zebrafish myocilin.

To disrupt this gene using the CRISPR/Cas9 genome editing, we designed a CRISPR RNA (crRNA) targeting exon 1 (Figure 1A). The ribonucleoprotein (RNP) complexes (crRNAs/trans-activating CRISPR RNA (tracrRNA) and Cas9 protein) were microinjected into the animal pole of AB zebrafish embryos at the one-cell stage of development (Figure 1B). The injected embryos (F0) were raised to adulthood and screened using polyacrylamide gel electrophoresis (PAGE) and Sanger sequencing for the presence of germline transmitted *myoc* deletions. Sanger sequencing identified mutant mosaic fishes transmitting an indel variant (c.236_239delinsAAAGGGGAAGGGGA), which was predicted to cause a frameshift in the coding region and a premature termination codon p.(V75EfsX60) (Figure 1D). These mosaics were selected as F0 founders and backcrossed with wild-type AB zebrafish to segregate off-targets, and the offspring (F1) was genotyped by PAGE (Figure 1B). Mutant F1 heterozygotes were outbred again with wild-type AB to further segregate off-target mutations in the F2 generation (Figure 1B). Finally, F3 homozygous *myoc* KO fish were obtained through inbreeding F2 heterozygotes. F3 genotyping by PAGE (Figure 1C) and Sanger sequencing (Figure 1D) showed agreement of the proportions of the three genotypes with the expected Mendelian ratios, indicating that *myoc* disruption does not affect zebrafish fertility and viability.

The mutation is predicted to result in degradation of the *myoc* mRNA by the non-sense mediated decay (NMD) pathway [49], leading to a loss-of-function (LoF) of the protein. To confirm this hypothesis, *myoc* mRNA levels were analysed by qRT-PCR in three pools of 15 larvae each per genotype. We observed a reduction of mRNA levels in heterozygous (+/−) and mutant homozygous (−/− or KO) larvae of approximately 50% and 80%, respectively, with respect to the value of their wild-type (+/+) littermates (Figure 1E). These results were in accordance with our hypothesis, and although there was a low level of residual mutant mRNA in the KO larvae, it would be likely translated into a non-functional truncated protein that contains only 74 normal amino acids. All these data support that the generated mutation produces a complete *myoc* LoF.

### 3.2. Expression of Myoc and Phenotypic Characterisation of the KO Zebrafish Line

To the best of our knowledge, myocilin expression has not been described in zebrafish. To investigate the presence of this protein in zebrafish by immunohistochemistry we used a chicken anti-myocilin antibody that was raised against a N-terminal peptide of the human protein (TNT antibody), and we took advantage of the KO zebrafish line as a unique negative control to assess the specificity of the signals. FWIHC confocal three-dimensional reconstruction of 96 hpf embryos revealed positive immunolabeling in the lens epithelium and intercellular spaces on both the external surface of the optic cup and dorsoposterior and ventral periocular tissues, which probably corresponds to the periocular mesenchyme (Figure 2A–C and Appendix A).

Positive immunolabeling was also detected apparently in the extracellular space of the yolk surface (Figure 3A–C and Appendix A) and in caudal muscular tissue (Figure 4A–C). 

These immunosignals were reduced in heterozygous embryos (Figure 2D–F and Figure 3D–F and Figure 4D–F), and they were undetectable in −/− embryos (Figure 2G–I and Figure 3G–I and Figure 4G–I), which supports their specificity. The absence of green signals in the control sections that were incubated with preimmune antibody also supported the specificity of the immunoreactivity (Appendix A). 

Next, we investigated the presence of myocilin in adult zebrafish ocular tissues using immunohistochemistry. We detected positive immunoreactivity in the non-pigmented ciliary epithelium, blood vessels and stroma of the iris (Figure 5A), corneal epithelium (Figure 5B), and retinal ganglion cells (Figure 5C,D). These signals were absent in −/− zebrafish (Figure 5E–G), as well as in samples treated with the preimmune antibody or blocked with the antigenic peptide (Appendix A). This also supports their specificity.

To investigate the presence of myocilin in representative non-ocular adult tissues we selected skeletal muscle (pharyngeal muscle) and the digestive (intestinal bulb, and middle intestine) and reproductive (testis and ovary) systems. Analysis of tissue sections revealed myocilin immunoreactivity in the periphery of pharyngeal muscular fibres (Figure 6A,B). In the dilated portion of the proximal intestine, i.e., the intestinal bulb, we observed immunolabeling in the enterocyte apical side (Figure 6C,D), and, in contrast, epithelial cells of the middle intestine were intracellularly stained (Figure 6E,F), showing distinctive brush border staining (Figure 6E,F, arrows). These immunosignals were almost absent in the −/− (Figure 6G–L), and in additional negative controls, i.e., the preimmune antibody (Appendix A) and the competitive assay with the immunising peptide (Appendix A).

For the reproductive system, positive immunolabeling was observed in the follicular epithelium of the ovary and cortical granules of vitellogenic oocytes (Figure 7A,B, yellow and white arrows, respectively), as well as likely in the seminiferous epithelium, but apparently not in sperm (Figure 7C,D). As will be described in the next section, the absence of −/− females precluded their use as negative immunohistochemistry controls. However, the immunosignal was absent in the testis of −/− animals (Figure 7E,F), and significantly reduced in tissue sections of ovaries that were either treated with the preimmune antibody (Figure 6A) or blocked with the antigenic peptide (Appendix A), indicating that the immunolabeling is specific.

Because of the relationship between *myocilin* and glaucoma, the eyes were the principal focus of our phenotypic analysis. External morphological examination of both embryo (96 hpf) and adult (7 months) *myoc* KO zebrafish did not reveal any significant difference between the eyes and head of wild-type and KO zebrafish (Appendix A). Similarly, comparison of hematoxylin–eosin stained tissue sections between +/+ and −/− zebrafish embryos (96 hpf) showed no evident head (Appendix A) or eye (Appendix A) differences. The dorsal and ventral anterior segment structures of adult (7 months) −/− zebrafish were also similar to those of +/+ animals (Appendix A), although, an apparently increased folding of the anterior retina was observed in mutant zebrafish (Appendix A, arrow), compared to wild-type animals (Appendix A.B). Finally, no gross alterations were observed in the retina of −/− zebrafish (Appendix A). These data show that under our experimental conditions, *myoc* KO zebrafish are phenotypically indistinguishable from wild-type animals. 

Throughout the breeding process we observed a consistent absence of females among −/− zebrafish. To confirm this observation, a total of eight different heterozygote sister–brother mating from three consecutive generations (F3 to F5) were performed. The offspring from each mating was raised to adulthood (3 months), genotyped using PAGE, and the male/female ratio corresponding to each genotype was calculated based on examination of multiple external dimorphic phenotypes, including body shape, anal fin coloration, and presence or absence of a genital papilla [50]. We observed a significant decrease in female proportion that correlated with the KO genotype, i.e., approximately 41% of +/+ animals were female, whereas this percentage was reduced to 25 and 0% in +/− and −/− zebrafish, respectively (Figure 8A). These results indicate that myocilin might play a key role in sex determination in zebrafish. As a control to evaluate the possible lethality that might be associated with the KO allele, we analysed genotype and survival proportions in the offspring of inbred *myoc* heterozygotes. The genotype proportions did not differ significantly from the expected Mendelian values (+/+ 25%, +/− 50% and −/− 25%, *p* > 0.05) (Figure 8B) and survival at 24 and 96 hpf was similar to that of embryos that were obtained from wild-type progenitors (*p* > 0.05) (Figure 8C). 

These data show that there is not lethality associated with the KO genotype, ruling out that the sex proportion observed in KO animals is biased by sex-dependent lethality that is associated with a specific mutant *myoc* genotype. Moreover, 11 randomly selected −/− male siblings were mated with +/+ females, and all of them fertilised eggs after spontaneous spawning, confirming the fertility and sex of these animals and ruling out errors in sex classification.

### 3.3. Histology and Terminal dUTP Nick-End Labeling (TUNEL) of the Immature Gonad of the Myoc KO Zebrafish Line

Consecutive tissue sections were employed for immunohistochemical, and TUNEL analyses of the juvenile ovary-to-testis-transforming zebrafish gonad (28 dpf) in serial cryosections from eight individuals (four +/+ and four −/−). Vasa immunostaining on tissue section one confirmed the presence of the juvenile gonad in the ventral side of the swim bladder and above the intestine of both +/+ (Figure 9A,B) and −/− (Figure 9E,F). Myocilin immunodetection on tissue section two revealed some positive cells in +/+ juvenile gonads (Figure 9C, arrows). As expected, myocilin immunoreactivity was absent in −/− gonadal tissue (Figure 9G), indicating that the signal was specific. Given that oocyte apoptosis participates in the mechanism of testicular and ovarian differentiation in the larval-juvenile zebrafish transition [51], we also investigated gonadal apoptosis in KO animals using a TUNEL assay on tissue section four. No positive cells were observed in none of the +/+ gonads obtained from four individuals (Figure 9D), indicating that they corresponded to juvenile females. In contrast, the germinal tissue of −/− animals showed many TUNEL positive primordial germ-like cells, characterized by large and round nuclei (Figure 9H, arrows). Small nucleated cells that were located in the outer intestinal wall, were also positive for TUNEL staining (Figure 9H, arrowheads). The absence of labelling in the respective negative controls (tissue sections four to six) supported the specificity of the results (Appendix A).

Preliminary histological examination of the juvenile ovary-to-testis-transforming gonad (28 dpf) by hematoxylin-eosin staining did not show significant differences between wild type and KO zebrafish (data not shown), indicating that myoc LoF does not alter gonadal development at this stage. Further studies are required to confirm this observation.

### 3.4. Comparison of Transcriptomic Profiles of Adult Male −/− and +/+ Zebrafish

To investigate differences in gene expression profiles associated with *myoc* LoF we performed comparative whole transcriptome sequencing of adult male *myoc* KO and adult male wild-type zebrafish (2.5 months). Mutant and wild-type zebrafish siblings were obtained by inbreeding heterozygous F4 progenitors. To minimize the effect of individual variability, we pooled 3 fishes in each sample and two independent biological replicas of each experimental group were analysed.

A total of 10,168 genes with zero counts across all samples were excluded from the study, leaving 29,819 genes for further analysis (the transcriptomic analysis identified coding RNAs and multiple non-coding polyadenylated RNAs). The similarity between samples, evaluated using Pearson’s coefficient, indicated that the replicas were similar (Appendix A). Hierarchical clustering analysis based on the criteria of fold change ≥2 and raw *p*-value <0.05 for the comparison male KO vs. male wild-type of differentially expressed gene (DEG) patterns in the experimental replicas revealed similar clusters of DEGs between the two replicas of each experimental group, indicating that most identified gene expression patterns were reproducible (Appendix A). 

We identified an average of 6184 genes significantly upregulated (fold change >2 and raw *p* < 0.05) in the four comparisons (i.e., KO1 vs. WT1, KO1 vs. WT2, KO2 vs. WT1 and KO2 vs. WT2). Similarly, an average of 6023 genes were significantly down regulated (fold change < −2 and raw *p* < 0.05) in the four comparisons (Appendix A). Genes overrepresented in different biological processes, were identified by a KEGG enrichment analysis. We selected the top-20 regulated pathways that can be altered (absolute fold change >2) in the absence of *myoc* activity (Appendix A). All pathways were significantly different in the four comparisons. Fourteen were metabolic related pathways and six pathways were related with genetic information processing (Appendix A). 

We selected the top-50 down- (Figure 10A and Appendix A) and upregulated (Figure 10B and Appendix A) genes for further analyses. Complementary gene ontology functional enrichment analysis in the group of top-50 up-regulated genes using ShinyGO [48] showed significant DEGs in several categories including biosynthetic and catabolic processes and regulation of metabolic processes (Appendix A), in accordance with the general KEGG analysis. In addition, categories such as developmental processes, anatomical structure development and reproduction were also significantly enriched in this group. On the other hand, the cluster of top-50 down regulated genes showed significant DEGs in categories mainly related with biosynthetic processes, reproduction, meiosis, SMAD signal transduction and cell cycle (Appendix A). To validate the transcriptome results by qRT-PCR we selected the two most upregulated (*tuba7l* and *irg1l*) and the two most downregulated (*plpp4* and *grik3*). In addition, nine selected DEGs with absolute fold change values greater than 2.0, were also re-assessed by qRT-PCR. This group included five representative genes known to play key roles in zebrafish sexual differentiation (*dmrt1*, *amh*, *sycp3*, *cyp11a1* and *star*) and four Wnt genes that were also underexpressed in the *myoc* KO transcriptome (*dkk1a*, *lef1*, *ctnnbip1* and *dvl3a*). The Wnt genes were selected because of the role of Wnt signalling in zebrafish sex differentiation and the proposed role of myocilin as a Wnt modulator. 

The qRT-PCR results (Figure 10C,D and Appendix A) showed a good correlation with the transcriptome data and confirmed most expression differences, except that detected for *grik3* (Figure 10A,D and Appendix A). The higher sensitivity of qRT-PCR compared with RNA-Seq and the technical differences between the two procedures may explain this discrepancy and the differences observed in absolute FC values between the two techniques. In addition, we found significant downregulated expression in the KO transcriptome of two additional relevant genes involved in these processes: *ctnnb2* (fold change: −3.6; *p*-value: 1.0 × 10^−11^), which encodes beta-catenin 2, and *cyp19a1* (fold change: −18.1; *p*-value: 2.5 × 10^−7^), that is proposed to be a key player in ovary differentiation [52]. As expected *myoc* expression was found significantly downregulated in the KO animals (fold change: −4.7; *p*-value: 1.2 × 10^−18^). 

## 4. Discussion

Over the past two decades myocilin research has provided insight into the structure, expression and association with glaucoma of this protein. However, we still lack a clear understanding of its biological function and how mutant myocilin underlies glaucoma pathogenesis. Thus, these issues remain challenging scientific questions. To advance our knowledge on myocilin physiology, we generated a null *myoc* genotype in the zebrafish and assessed the resulting phenotypes. The obtained KO line is also a valuable control to investigate the specific histological localisation of myocilin by immunohistochemistry. The lack of well characterised anti-myocilin antibodies and suitable controls has hindered immunodetection of this protein and has provided controversial reports on myocilin expression. Our immunohistochemical analyses, which to the best of our knowledge, constitute the first report on myocilin expression in zebrafish, revealed the specific presence of myocilin in the eye and caudal muscles of zebrafish embryos (96 hpf). Myocilin was also detected in the secretory NPE cells of the adult zebrafish eye, mimicking the expression of the human gene [8,24]. These results also suggest that, similar to its human counterpart, the zebrafish protein may be secreted into the aqueous humour [10,11]. In addition, zebrafish myocilin was detected in the iris stroma, the corneal endothelium and the retinal ganglion cell layer, again resembling the localisation of the human protein [8]. However, and in contrast to humans, myocilin was almost undetectable in the zebrafish corneal epithelium [8]. Overall, parallels in myocilin expression between these two organisms indicate that zebrafish can be used as a model to study the role of myocilin in ocular physiology. In addition, myocilin expression was observed surrounding skeletal muscle cells in pharyngeal muscles, and intracellularly in epithelial cells of the middle intestine, with intense localisation in the brush border. 

Consistent with our results, expression of myocilin at the mRNA level has been detected in the human skeletal muscle [6,53] and small intestine [54]. It is interesting that the myocilin-related protein olfactomedin-4, is present in intestinal stem cells and is apparently involved in mucosal defence [55]. These results suggest that myocilin, as a secreted protein, might play a role in the muscular and small intestinal extracellular matrix, although further work is required to reveal the role of myocilin in these tissues. The presence of myocilin in adult male and female gonads (i.e., follicular epithelium, cortical granules of vitellogenic oocytes, and seminiferous tubule cells) indicates that it may play a role in gamete production in zebrafish, which is a hitherto unsuspected function. In addition, the expression of myocilin in undifferentiated germinal tissue suggests that this protein may play a role in sex differentiation as will be discussed later. 

Phenotype evaluation of the KO zebrafish showed neither gross macroscopic nor histological alterations in different organs that express *myoc*, such as the eye, skeletal muscle, brain, and digestive apparatus. These results indicate that *myoc* is not essential for normal body and tissue morphology, at least under normal conditions. Consistent with this conclusion, a previous report found that mouse *Myoc* is not required for either viability or for normal ocular morphology [56]. Moreover, and consistent with these results, it has been described that the homozygous premature termination codon p.(Arg46X) of human *MYOC*, a likely null variant, did not result in any identifiable pathogenic phenotype in a 77-year-old woman [57]. The apparently normal phenotype that is associated with *MYOC* LoF in different species, may result from a functional redundancy of other related gene products, which, at least under normal conditions, might balance the functional lack of myocilin in most *myoc*-expressing tissues. However, one striking finding of our study was the lack of KO female zebrafish, suggesting that this gene is required for female zebrafish sex determination. Known environmental factors that influence zebrafish sex, such as temperature, density, hormones, food and hypoxia [34], were controlled and homogeneous in our experimental crosses, making it highly unlikely that they biased the observed sex ratio. Moreover, none of the several *loci* identified in zebrafish that appeared to influence sex ratios in a strain-dependent manner [58,59,60,61], were located on the *myoc locus*, at chromosome 20. Thus, these data indicate that *myoc* may be a novel gene that is involved in zebrafish sex determination. This idea is supported by the expression of myocilin in the immature juvenile gonad (28 dpf). In addition, the fact that the absence of this protein in the KO zebrafish results in only males with normal testis, indicates that this protein is required for ovary differentiation, although the presence of myocilin in wild-type zebrafish does not block ovary development in all individuals, suggesting that additional factors might modulate the role of myocilin as a pro-female factor. Further studies, including detailed histological, immunohistochemical and gene expression analyses of both primordial germ cells and juvenile ovary-to-testis-transforming gonads are required to elucidate the mechanism underlying myocilin’s role in gonad differentiation.

To characterize the KO phenotype at the gene expression level we performed a transcriptomic analysis, which revealed many DEGs, indicating the existence of marked differences between adult males, associated with the *myoc* KO genotype. The top-20 significant KEGG pathways were grouped into two broad functional categories, metabolic and genetic information processing-related. This likely identifies broad molecular processes involved in the male-biased sex ratio observed in the KO animals. Interestingly, detailed functional enrichment evaluation, focused on the top-50-up and -down DEGs, identified significant categories related with reproduction, meiosis, anatomical structure and development, indicating that these processes participate in the male-biased proportion associated with the null *myoc* genotype. Alpha-tubulin genes *tuba7l* and *tuba4l* were the most over- and under-expressed genes in *myoc* KO males, respectively. In accordance with our results, *tuba7l* is overrepresented in testes of sexually mature zebrafish [62] while *tuba4l* is the most downregulated gene in male gonads of zebrafish masculinized by treatment with high temperature [62,63].

Interestingly, we identified several highly DEGs in male zebrafish *myoc* KO that play key roles in male sexual differentiation, including *dmrt1*, *amh*, *sycp3*, *cyp11a1, cyp11c1* and *star*. *Dmrt1* regulates *amh* and *foxl2* and is necessary for zebrafish male sexual development [64]. The *amh* gene encodes the anti-Müllerian hormone, a member of the TGF-beta superfamily of growth factors. This hormone is upregulated in testis of sexually mature zebrafish [62] and promotes male development, although is not essential for this process [65]. *Sycp* is a pro-male gene, and a spermatocyte marker for meiotic cells [66]. *Cyp11a1* and *star* are two steroidogenic genes encoding rate limiting steps in steroid biogenesis androgen production [67,68]. In addition, *cyp11c1* is required for juvenile ovary-to-testis transition, Leydig cell development, and spermatogenesis in males because it participates in the synthesis of the major natural androgen in teleost fish (11-ketotestosterone) [69,70]. In accordance with our results, some of these genes have also been described to be upregulated (*dmrt1, amh, sycp3, cyp11c1* and *star*) or downregulated (*cyp11a1*) in male gonads of zebrafish masculinized by treatment with elevated water temperature (35 °C) [63]. Overall, these data reveal that *myoc* LoF results in over-expression of genes required for male development.

Remarkably, and in accordance with the role of the canonical Wnt/beta-catenin signalling as a pro-female pathway [38], we found that several Wnt signalling genes (*dvl3a*, *lef1, ctnnb2* and *ctnnbip1*) were under-expressed in KO male zebrafish. *Dvl3a* gene product is required for activation of zygotic Wnt/beta-catenin signalling and the Wnt/planar cell polarity pathway [71]. The transcription factor *left1* associates with beta-catenin to activate gene expression [72] and *ctnnb2* encodes beta-catenin 2, which is the key transcription factor of this signalling pathway. Under-expression of *ctnnbip1*, a repressor of beta-catenin-TCF-4-mediated transactivation [73], might contribute to balance Wnt pathway attenuation, although the actual meaning of this expression change remains to be investigated. In addition, the Wnt secreted inhibitor *dkk1a* [74] was overexpressed, which may contribute to Wnt signalling depression. In accordance with our results, it has been reported that myocilin modulates the Wnt signalling pathway [29,30] and that it interacts with two secreted inhibitors of Wnt signalling (Frizzled-related proteins 1 and 3), and with various Frizzled receptors (Fzd1, Fzd7 and Fzd10) [29]. On the other hand, inhibition of Wnt/beta-catenin signalling results in an increased proportion of zebrafish males and downregulated expression of *cyp19a1a* and *lef1* genes [38]. Overall, these data support that *myoc* LoF downregulates the Wnt signalling pro-female pathway [38,75], inhibiting ovary differentiation and leading to testis formation. Moreover, it is known that beta-catenin is an essential transcriptional regulator of aromatase [76], a gene proposed to play a key role in directing ovarian differentiation and development [52]. In accordance with this fact, *cyp19a1* was also under-expressed in the KO zebrafish transcriptome. Therefore, decreased expression of both the Wnt signalling pathway and aromatase (*cyp19a1*) associated with myocilin LoF may block ovary differentiation of the juvenile zebrafish gonad, leading to testis development. Wnt signalling also plays a role in sex determination in other teleosts [77,78] and mammals [79], thus the possible role of myocilin in mammal and human sex determination remains to be investigated. 

## 5. Conclusions

Our results show that the established *myoc* KO zebrafish line is a useful model to investigate the elusive function of this protein, mainly known for its association with glaucoma. In addition, this study provides evidence for a previously unsuspected role of myocilin as a novel protein required for ovary differentiation associated with downregulated gene expression of the Wnt signalling pathway.

## Figures and Tables

**Figure 1 biology-10-00098-f001:**
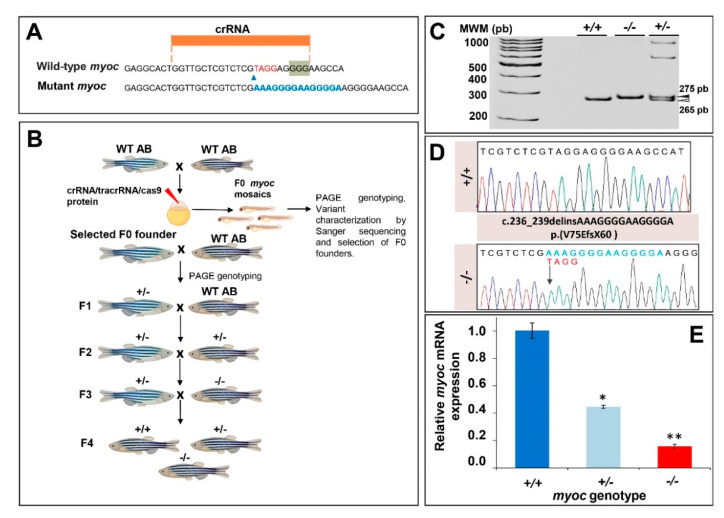
Generation and molecular characterisation of a *myoc* KO zebrafish line using CRISPR/Cas9 genome editing. (**A**) Localisation of the crRNA designed to target *myoc* exon 1. The PAM sequence is indicated on a green background. Red and blue nucleotides indicate deleted and inserted nucleotides identified in mutant fishes used to establish the KO line, respectively. (**B**) Stepwise procedure that was followed to establish the KO line. F0 animals were raised to adulthood, crossed with wild-type AB zebrafish and the offspring was genotyped by PAGE to identify germline transmission of *myoc* deletions (F0 founders). F0 founders were crossed with wild-type AB animals to obtain mutant F1 heterozygotes that were further outbred to segregate off-targets mutations and to obtain the F2 generation. F2 heterozygotes were inbred to produce the F3 generaton. The Biorender tool was used to create this scheme. (**C**) Genotyping of a *myoc* indel using 8% PAGE. Three representative samples are shown. White arrowhead: wild-type allele (265 bp). Black arrowhead: mutant allele (275 bp). (**D**) Sanger sequencing of the selected *myoc* mutation. The arrow in the electropherogram indicate the nucleotide where the mutation starts. Blue and red letters indicate inserted (14 bp) and deleted (4 bp) nucleotides, respectively. (**E**) Decreased *myoc* mRNA levels in *myoc* mutant zebrafish (+/− and −/−). mRNA levels in pools of 15 F3 zebrafish larvae (144 hpf) were measured by qRT-PCR. The results are expressed as relative expression levels and normalised to control wild-type (+/+) larvae. Values represent the average of three different experiments. Asterisks indicate statistical significance compared to +/+. * *p* = 00051. ** *p* =0.00017.

**Figure 2 biology-10-00098-f002:**
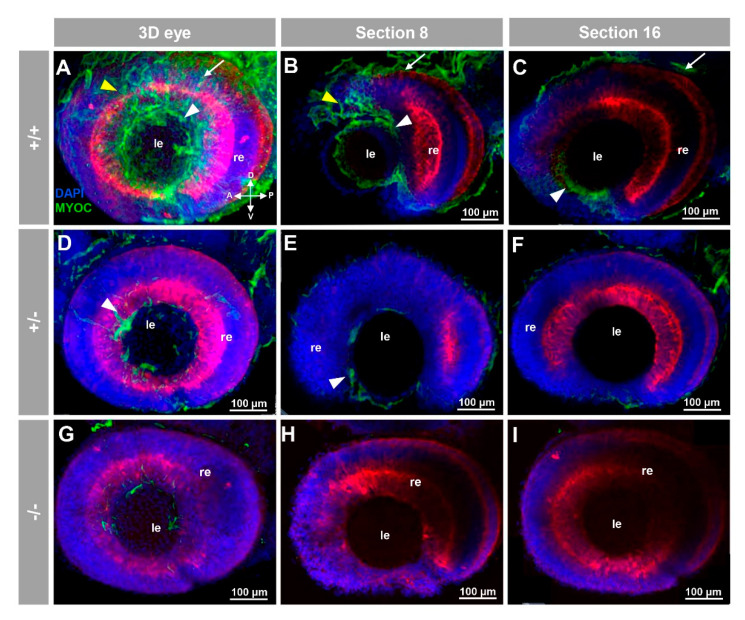
Fluorescent whole-mount immunohistochemical detection of myocilin in the eye of zebrafish embryos (96hpf). Wild-type (**A**–**C**), heterozygous (**D**–**F**) and homozygous (**G**–**I**) *myoc* mutant embryos were incubated with a chicken conjugated goat anti-chicken IgY secondary antibody. Three-dimensional reconstruction from z-stack scanned confocal microscopy images (**A**,**D**,**G**) of the eye. Optical sections (92 μm) 8 (**B**,**E**,**H**) and 16 (**C**,**F**,**I**), from the exterior ocular surface, were selected from z-stack images to show the precise localisation of the green signal in the external and internal surface of the optic cup (white arrows and yellow arrowheads, respectively), lens epithelium (white arrowhead), and dorsoposterior and ventral periocular tissues (yellow arrowheads) (**A**–**C**). Blue: DAPI nuclear staining. Green: Cy2-conjugated goat anti-chicken IgY secondary antibody. Red: tissue autofluorescence. The cross indicates the position of the embryonic axes (D: dorsal; P: posterior; V: ventral; A: anterior). The images are representative of the result observed in 10 embryos. le: lens. re: retina. The negative controls are shown in Appendix A. Two-dimensional confocal image z-stacks are shown in Appendix A.

**Figure 3 biology-10-00098-f003:**
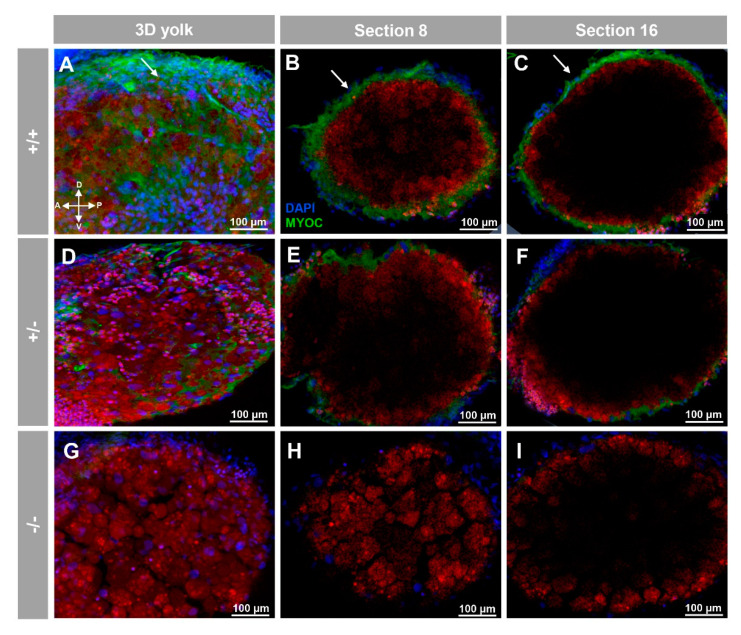
Fluorescent whole-mount immunohistochemical detection of myoc in the yolk of zebrafish larvae (96hpf). Wild-type (**A**–**C**), heterozygous (**D**–**F**) and homozygous (**G**–**I**) *myoc* mutant embryos were incubated with chicken anti-myocilin (TNT) primary antibody and a Cy2-conjugated goat anti-chicken IgY secondary antibody. Three-dimensional reconstruction from z-stack scanned confocal microscopy images (**A**,**D**,**G**) of the yolk. Sections (92 μm) 8 (**B**,**E**,**H**) and 16 (**C**,**F**,**I**), were selected from z-stack images to show the precise localisation of the green signal on the yolk’s surface (arrows) (**A**–**C**). Blue: DAPI nuclear staining. Green: Cy2-conjugated goat anti-chicken IgY secondary antibody. Red: tissue autofluorescence. The cross indicates the position of the embryonic axes (D: dorsal; P: posterior; V: ventral; A: anterior). The images are representative of the result observed in 10 embryos. The negative controls are shown in Appendix A. Two-dimensional confocal image z-stacks are show in Appendix A.

**Figure 4 biology-10-00098-f004:**
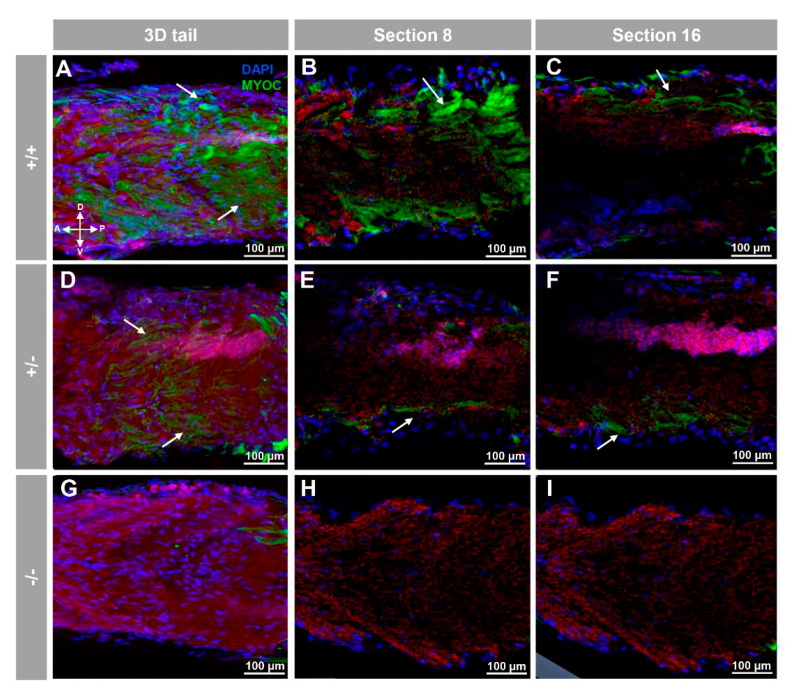
Fluorescent whole-mount immunohistochemical detection of myoc in the tail of zebrafish embryos (96hpf). Wild-type (**A**–**C**), heterozygous (**D**–**F**) and homozygous (**G**–**I**) *myoc* mutant embryos were incubated with a chicken anti-myocilin (TNT) primary antibody and a Cy2-conjugated goat anti-chicken IgY secondary antibody. Three-dimensional reconstruction from z-stack scanned confocal microscopy images (**A**,**D**,**G**) of the tail. Sections (92 μm) 8 (**B**,**E**,**H**) and 16 (**C**,**F**,**I**) were selected from z-stack images to show the precise localisation of the green signal in the tail’s skin (arrow) (**A**–**C**). Blue: DAPI nuclear staining. Green: Cy2-conjugated goat anti-chicken IgY secondary antibody. Red: tissue autofluorescence. The cross indicates the position of the embryonic axes (D: dorsal; P: posterior; V: ventral; A: anterior). The images are representative of the results observed in ten embryos. The negative controls are shown in Appendix A.

**Figure 5 biology-10-00098-f005:**
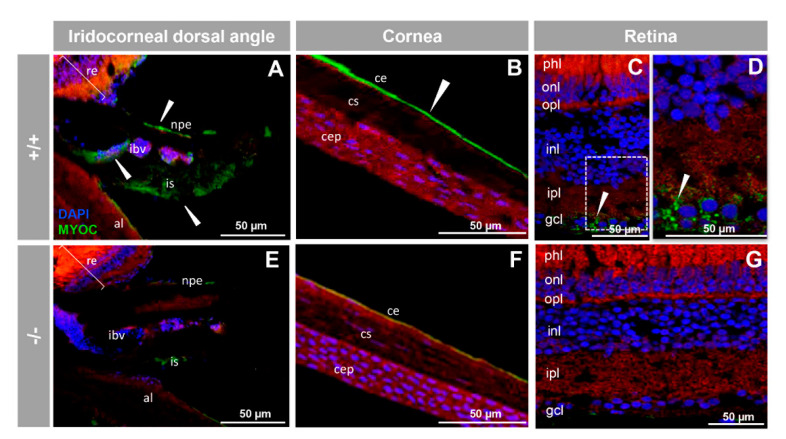
Immunohistochemistry of myocilin in ocular structures of adult zebrafish. Fluorescent immunohistochemistry of iridocorneal dorsal angle (**A**,**E**), cornea (**B**,**F**) and retina (**C**,**D**,**G**) sections (14 μm) of wild-type and KO *myoc* adult zebrafish (7 months). Samples were incubated with an anti-myocilin primary antibody (TNT), followed by Cy2-conjugate goat anti-chicken IgY secondary antibody (green signals). The TNT antibody recognises the N-terminal part of myocilin protein. Expression is seen in the non-pigmented epithelium of the ciliary body and in the iris stroma and iris vessels (**A**), in the corneal endothelium (**B**), and in the ganglion cell layer in the retina (**C**,**D**) (arrowhead) in wild-type. Red signals correspond to tissue autofluorescence and blue signals correspond to DAPI nuclear staining. al: annular ligament. npe: non-pigmented ciliary epithelium; ibv: iris blood vessels; is: iris stroma. cep: corneal epithelium; cs: corneal stroma; ce: corneal endothelium glc: ganglion cell layer; ipl: inner plexiform layer; inl; inner nuclear layer; opl: outer plexiform layer; onl: outer nuclear layer; phl: photoreceptor layer. re: retina+/+: wild-type; −/−: *myoc* KO. The negative controls are shown in Appendix A.

**Figure 6 biology-10-00098-f006:**
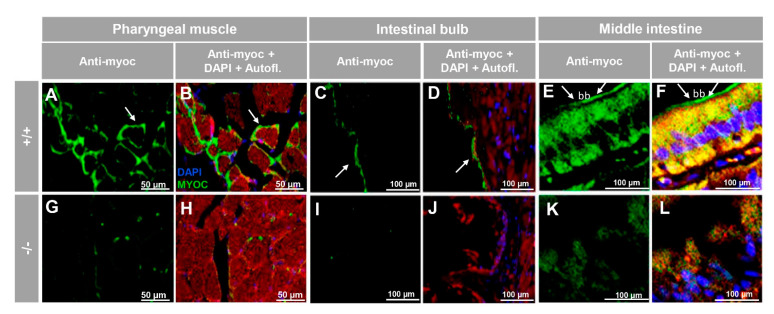
Immunohistochemistry of myocilin in non-ocular tissues of adult zebrafish. Wild-type and *myoc* KO adult (7 months) zebrafish tissue sections (14 μm) were incubated with a chicken anti-myocilin (TNT) primary antibody and a Cy2-conjugate goat anti-chicken IgY secondary antibody. Arrows show immunostaining in the periphery of pharyngeal muscular fibres (**A**,**B**,**G**,**H**), the enterocyte apical side in the intestinal bulb (**C**,**D**,**I**,**J**) and in the brush border of the epithelial cells in the middle intestine (**E**,**F**,**K**,**L**). Red signals correspond to tissue autofluorescence. bb: brush border. The images are representative of the results observed in three tissue sections from three animals. +/+: wild-type; −/−: *myoc* KO. Negative controls are shown in Appendix A.

**Figure 7 biology-10-00098-f007:**
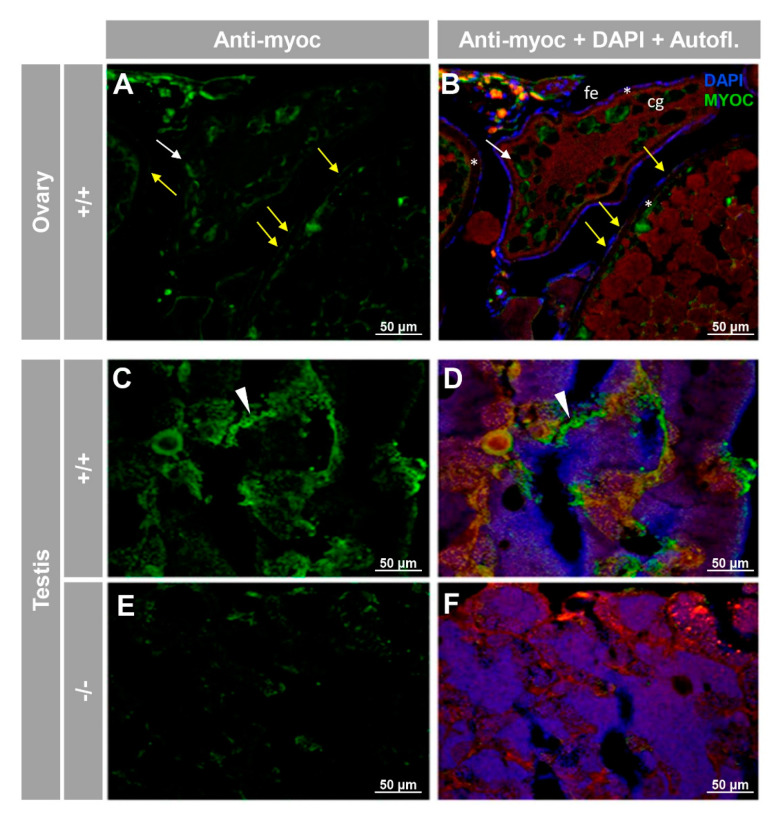
Immunohistochemical detection of myoc in the reproductive system of adult zebrafish. Tissue sections (14 μm) of adult (7 months) wild-type ovary (**A**,**B**) and testis (**C**,**D**) and KO *myoc* testis (**E**,**F**) were incubated with an anti-myocilin (TNT) primary antibody, followed by Cy2-conjugate goat anti-chicken IgY secondary antibody (green signals). Note that ovaries from −/− animals were not available because of the absence of females with this genotype. White and yellow arrows indicate cortical granules and follicular epithelium-associated immunoreactivity, respectively. White arrowheads show immunolabeling in the seminiferous epithelium (**C**,**D**). Red signals correspond to tissue autofluorescence. The images are representative of the result observed in three tissue sections from three animals. +/+: wild-type; −/−: myoc KO. cg: cortical granules. fe: follicular epithelium. *: zona radiata. Negative controls are shown in Appendix A.

**Figure 8 biology-10-00098-f008:**
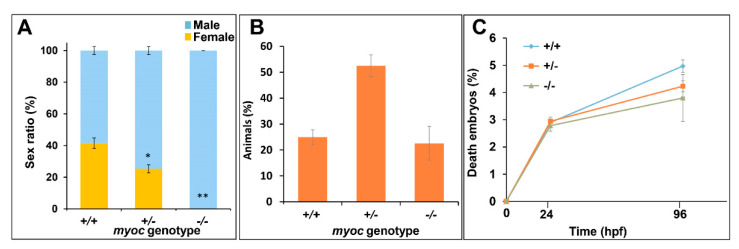
Absence of females among myoc KO zebrafish. Eight +/− *myoc* siblings were mated, and 30–35 embryos per cross were randomly selected (n = 256), raised to adulthood and genotyped by PAGE. The sex ratio as a function of the *myoc* genotype was calculated (**A**). Observed genotype proportions (**B**) and embryo survival (**C**). Values are expressed as the mean ± standard error of the mean (SEM). Asterisks indicate statistical significance compared to +/+. * *p* = 0.00038. ** *p* = 0.00001. +/+: wild-type; −/−: *myoc* KO.

**Figure 9 biology-10-00098-f009:**
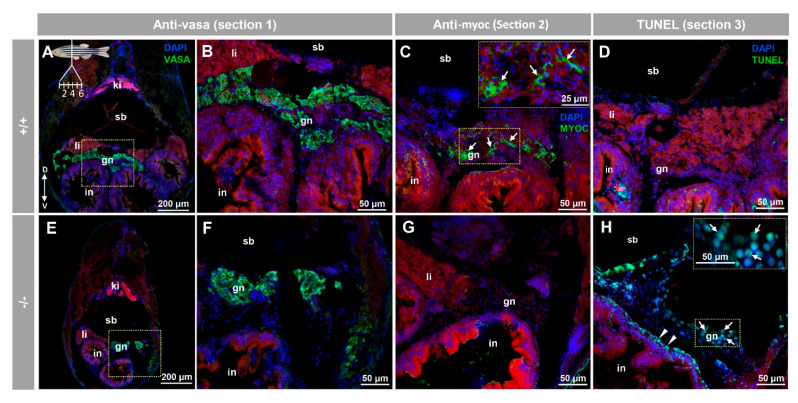
Myocilin expression and apoptosis analysis in the immature (28 dpf) gonad of zebrafish KO for *myoc*. Analyses were performed on three consecutive tissue sections (1 to 3, upper left insert in panel **A**). Vasa immunolabeling was carried out on section one (green, **A**,**B**,**E**,**F**). Areas indicated by yellow rectangles in (**A**,**E**) are magnified in (**B**,**F**), respectively. Tissue section two was incubated with an anti-myocilin (TNT) antibody (green signals) to localise myocilin expression (**C**,**G**). Apoptosis was assessed in tissue section three using terminal dUTP nick-end labeling (TUNEL) of fragmented DNA (**D**,**H**). Arrows in (**C**) indicate myocilin immunoreactivity. Arrows and arrowheads in H indicate TUNEL-positive cells in gonadal tissue and in the outer intestinal wall, respectively. Blue and red signals correspond to DAPI nuclear staining and tissue autofluorescence, respectively. gn: gonad. in: intestine. ki: kidney. li: liver. sb: swimbladder. The vertical double arrow in (**A**) indicate de dorsovental axis (D: dorsal; V: ventral). The images are representative of the results observed in four fishes of each genotype. Negative controls of the juvenile gonad are shown in Appendix A (tissue sections four to six).

**Figure 10 biology-10-00098-f010:**
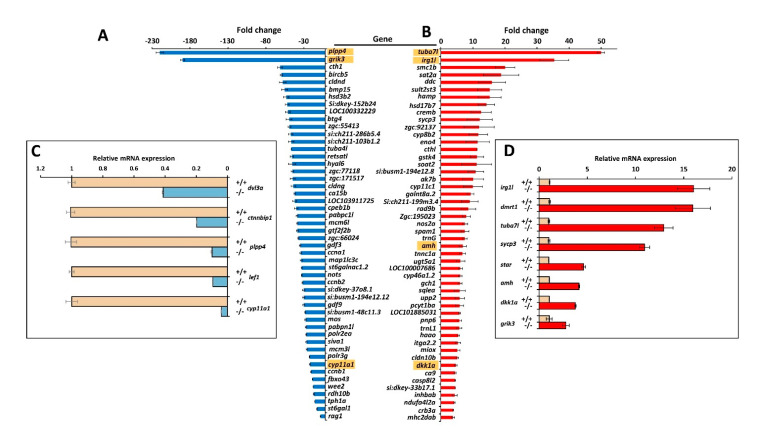
Top-50 DEGs in *myoc* KO male versus wild-type male zebrafish. Down- (**A**) and upregulated (**B**) genes identified by high throughput RNA sequencing with significant differences in the four comparisons (KO1 vs. WT1, KO1 vs. WT2, KO2 vs. WT1 and KO2 vs. WT2). The golden background indicates genes that were analysed by qRT-PCR. Confirmation by qRT-PCR of differential gene expression of selected down- (**C**) and upregulated (**D**) genes.

## Data Availability

The data presented in this study are available in the Appendix A.

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
