# Peer review of "Knockout of myoc Provides Evidence for the Role of Myocilin in Zebrafish Sex Determination Associated with Wnt Signalling Downregulation"

_biology, 2021, doi:10.3390/biology10020098_

Round 1

Reviewer 1 Report

The manuscript entitled "Knockout of myoc reveals the role of myocilin in zebrafish sex determination associated with Wnt signaling downregulation" is of interest to sex determination study. Although the knockout of myoc did not cause any apparent gross morphological anomaly in other tissues, it was more focused on myocilin expression and function in the eye, other tissues rather than the gonad. It is this reviewer's opinion that the content of this manuscript did not meet the expectation of the title. The authors will need to revise the title or provide data to support the title. Besides, more information related to sex determination in zebrafish should be provided in the introduction section. A paper entitled "Sex Reversal in Zebrafish fancl Mutants Is Caused by Tp53-Mediated Germ Cell Apoptosis" from Dr. Postlethwait's group can serve as a good guide for this study.

Author Response

Response to reviewer 1 comments.

Reviewer’s comment: “It is this reviewer's opinion that the content of this manuscript did not meet the expectation of the title. The authors will need to revise the title or provide data to support the title.”

Response: According to this comment we have changed the title as follows: “Knockout of myoc provides evidence for the role of myocilin in zebrafish sex determination associated with Wnt signalling downregulation”.

Reviewer’s comment: “Besides, more information related to sex determination in zebrafish should be provided in the introduction section. A paper entitled "Sex Reversal in Zebrafish fancl Mutants Is Caused by Tp53-Mediated Germ Cell Apoptosis" from Dr. Postlethwait's group can serve as a good guide for this study.”

Response: Thank you for the comment and for the reference. We have completed the introduction as suggested. To keep manuscript length reasonable, some paragraphs on sex determination included in the discussion have been moved to the end of the introduction (lines 101-113).

Reviewer 2 Report

The authors present a straight-forward analysis of myocilin in zebrafish.  Myocilin has been implicated in glaucoma and they generated a knockout zebrafish line using CRIPSR/CAS9 to try to elucidate the mechanism of action.  They generated an allele that produced an early stop codon as well as nonsense mediated decay of the mRNA. They then used a published antibody to examine the expression of myoc in the eye, yolk, zebrafish tail, pharyngeal muscle, intestinal blub, middle intestine and reproductive tissue.  Mutant survival was normal and they observed no other obvious defects.  The authors observed the surprising observation that homozygotic knockouts were exclusively male.  The authors then took a transcriptomics approach and identified up and downregulated genes in mutant embryos and observed several Wnt genes and 5 genes known to be involved in sexual differentiation. 

Overall the authors present a novel discovery and all of the experiments are carefully controlled.  I have no major concerns, and I think this work will be of interested and should be accepted.

Minor points:

1. At multiple points in the summary, abstract and introduction the authors use the phrase " known for its association with glaucoma".  This is an important point for the paper but is written in a somewhat vague way.  In the discussion they use the more precise language " we lack a clear understanding of its biological function and how mutant myocilin underlies glaucoma pathogenesis".  It would be clearer to the reader if this was stated in the abstract or introduction in a similar way.  

Author Response

Response to reviewer 2 comments.

Reviewer’s comments: “At multiple points in the summary, abstract and introduction the authors use the phrase " known for its association with glaucoma". This is an important point for the paper but is written in a somewhat vague way. In the discussion they use the more precise language " we lack a clear understanding of its biological function and how mutant myocilin underlies glaucoma pathogenesis". It would be clearer to the reader if this was stated in the abstract or introduction in a similar way.”

Response: Thank you very much for your comments. Following your recommendations, we have made changes in the Simple Summary (line 18), Abstract (line 28) and Introduction (lines 47-48). We hope you find these modifications acceptable.

Reviewer 3 Report

The study used CRISPR/Cas9 technology to generate myocilin knockout zebrafish and found it will affect sex determination especially the female gonad formation. The author also found the homozygous myoc mutant male can mate with sibling wild type female and got the fertilized eggs. The immature gonad in homozygous myoc mutant contained many apoptotic cells. The expression of ctnnb2 and cyp19a1 required for ovary differentiation were decreased in the mutant. It is an interesting mutant phenotype and I have some questions:

Major comment

  1. The author should provide evidence for cell death in the ovary.
  2. The author should provide histological evidence that defects in ovary maturation in the mutant.

Minor comment

  1. In figure 9D, SB should be “sb”.
  2. In figure S6, why there was still some myoc staining in the cortical granules (negative control)?

Author Response

Response to reviewer 3 comments.

Reviewer’s comment: “The author should provide evidence for cell death in the ovary.”

Response: Thank you for this comment, but please note that the ovary of the myoc KO zebrafish line cannot be analysed because all adult animals are males. We have evaluated cell death in the immature gonad, showing the presence of apoptotic primordial germ-like cells (Figure 9H), which is compatible with normal testis differentiation.

Reviewer’s comment: “The author should provide histological evidence that defects in ovary maturation in the mutant.”

Response: As mentioned in the previous response, ovary maturation cannot be observed in this zebrafish line. Our immunohistochemistry data indicate that the process of gonad differentiation is normal with no evident abnormalities (Figure 7E), and results in fertile males (lines 490-492).

Reviewer’s comment: “In figure 9D, SB should be “sb”.”

Response: Thank you for this observation. We believe that the reviewer means “9C” instead of “9D”. We have changed “SB” to “sb” in figure 9C.

Reviewer’s comment: “In figure S6, why there was still some myoc staining in the cortical granules (negative control)?”

Response: The tiny green signal in Figure S6A may correspond to some non-specific immunoreactivity present in the preimmune antibody, which is clearly lower than that of the anti-myocilin antibody, as can be seen in Figure 7A,B. On the other hand, the green fluorescence observed in Figure S6B, which is a little bit higher than that of Figure S6A, may be the result of incomplete antibody blocking by the immunizing peptide in the competitive assay. This was described in the results as follows (lines 426-429): “the immunosignal was absent in the testis of -/- animals (Figures 7E,F), and significantly reduced in tissue sections of ovaries that were either treated with the pre-immune antibody (Figure 6A) or blocked with the antigenic peptide (Figure S6B)”.

Round 2

Reviewer 1 Report

This reviewer's concerns have been addressed.

Author Response

Thank you very much for your constructive comments and for your time.

Reviewer 3 Report

  1. The author provided the presence of apoptotic primordial germ-like cell (Figure 9H) in the myoc homozygous mutant and found they are all male and are fertile. Although there are a functional analysis and the statistic of the sex reversal. The detail of structure change in gonad is still absent. According to Ge’s studies, gonad will be remained in the ovary like before 25 dpf (Chen and Ge, 2013). In Postlewwait’s study, the sex reversal can be detected in the lack of apparance of enlarged oocytes between transitional juveniles (26 dpf) to post-transitional juveniles (32 dpf)(Rodríguez-Marí et al., 2010). Therefore, it would be better to see if the maturation of the oocyte from early to late-stage IB perinucleolar oocytes is affected by HE staining in the myoc mutant. It is also better to show the apoptotic cells (by TUNEL assay) are detected in the oocyte primordium in the immature gonad in the myoc mutant.

Reference

Chen, W., Ge, W., 2013. Gonad differentiation and puberty onset in the zebrafish: evidence for the dependence of puberty onset on body growth but not age in females. Mol Reprod Dev 80, 384-392.

Rodríguez-Marí, A., Cañestro, C., Bremiller, R.A., Nguyen-Johnson, A., Asakawa, K., Kawakami, K., Postlethwait, J.H., 2010. Sex reversal in zebrafish fancl mutants is caused by Tp53-mediated germ cell apoptosis. PLoS Genet 6, e1001034.

Author Response

Reviewer’s comment: “The detail of structure change in gonad is still absent. According to Ge’s studies, gonad will be remained in the ovary like before 25 dpf (Chen and Ge, 2013). In Postlewwait’s study, the sex reversal can be detected in the lack of apparance of enlarged oocytes between transitional juveniles (26 dpf) to post-transitional juveniles (32 dpf)(Rodríguez-Marí et al., 2010). Therefore, it would be better to see if the maturation of the oocyte from early to late-stage IB perinucleolar oocytes is affected by HE staining in the myoc mutant. It is also better to show the apoptotic cells (by TUNEL assay) are detected in the oocyte primordium in the immature gonad in the myoc mutant.”

Response: Thank you very much for these comments. We conducted a preliminary examination of the histology of the juvenile ovary-to-testis-transforming gonad (28 dpf) by hematoxylin-eosin staining, and we did not observe significant differences between wild type and knockout animals, indicating that gonad development in myoc mutants is normal at this stage. However, because the histological structure of the ovary-to-testis-transforming gonad is difficult to analyse simply by hematoxilin-eosin staining, these results must be confirmed in further investigations using, for instance, markers such as cyp19a1a and amh. These detailed work is beyond the main scope of the present study and it will be performed soon. In an effort to address the reviewer’s comment, we have modified the results section as follows (lines 520-524): “Preliminary histological examination of the juvenile ovary-to-testis-transforming gonad (28 dpf) by hematoxylin-eosin staining did not show significant differences between wild type and KO zebrafish (data not shown), indicating that myoc LoF do not alter gonadal development at this stage. Further studies are required to confirm this observation.” Please, note that we have also modified the discussion accordingly (lines 645-648): “Further studies, including detailed histological, immunohistochemical and gene expression analyses of both primordial germ cells and juvenile ovary-to-testis-transforming gonad are required to elucidate the mechanism underlying myocilin’s role in gonad differentiation”. Regarding apoptosis, we find difficult to evaluate apoptosis in the immature gonad using simple hematoxilin-eosin staining. In our opinion the results shown in Figure 9, where we analysed the germ cell marker vasa and TUNEL-positive cells in consecutive tissue sections, strongly support the presence of apoptotic cells in the immature gonad. We hope that you find these modifications satisfactory.